# Spatial 3D-LLM: Progressive Spatial Awareness for Advanced 3D Vision-Language Understanding

## Abstract

New era has unlocked exciting possibilities for extending Large Language Models (LLMs) to tackle 3D vision-language tasks. However, most existing 3D Multimodal LLMs (MLLMs) rely on holistic 3D scene information or specifically designated regions for 3D vision-language tasks, failing to capture multi-level location-based information. Addressing these concerns, we present Spatial 3D-LLM, a 3D MLLM specifically designed to enhance spatial perception and reasoning for 3D vision-language tasks by enriching the spatial embeddings of 3D scenes. **Spatial 3D-LLM** incorporates an LLM backbone and a meticulously designed progressive spatial awareness scheme that captures spatial information as the perception field expands, generating location-enriched 3D scene embeddings that serve as visual prompt. Additionally, we introduce two novel tasks, namely 3D object distance measurement and 3D layout editing, and construct a 3D instruction dataset **MODEL**, to inspire more profound 3D spatial perception capabilities. Experimental results demonstrate that Spatial 3D-LLM achieves state-of-the-art performance across a wide range of 3D vision-language tasks, revealing the improvements stemmed from our progressive spatial awareness scheme of mining more profound spatial information and the proposed dataset.

## 1 Introduction

In recent years, Vision-Language Models (VLMs)(Hong et al., 2023; Liu et al., 2024; Zhang et al., 2024; Wang et al., 2024) have rapidly advanced, with 2D Multimodal Large Language Models (MLLMs) demonstrating remarkable capabilities in understanding complex visual scenes(Li et al., 2022; Han et al., 2023). Concurrently, much success of developing 3D MLLMs has been achieved on 3D scene understanding(Hong et al., 2023; Guo et al., 2023; Wang et al., 2023). 3D spatial awareness encompasses the perception of spatial states, including locations and distances, as well as spatial reasoning and generation derived from this perception, such as embodied planning and spatial layout editing. While diving into 3D world, 3D spatial awareness is one of the keys for 3D MLLMs to perform downstream tasks, such as robotics(Gao et al., 2024), virtual and augmented reality(Konenkov et al., 2024) and interior design(Yang et al., 2024b).

To enable VLMs to perceive and comprehend the 3D world, most existing 3D MLLM architectures incorporate a 3D vision encoder to extract 3D features and align them with an LLM(Hong et al., 2023; Wang et al., 2023; Guo et al., 2023; Li et al., 2024). However, current methods(Chen et al., 2023b; Huang et al., 2023; 2024) primarily focus on object attributes, overlooking strategies for precise 3D location perception. Approaches like Hong et al. (2023), Zhu et al. (2024) and Chen et al. (2024b) utilize the Q-former(Li et al., 2022) module to extract instruction-related information from 3D scene embeddings to form 3D MLLM input. The extracted input embeddings are too correlated with the instructions to fully capture the spatial concepts of 3D scenes. Existing works (Wang et al., 2023; Hong et al., 2023; Li et al., 2024; Chen et al., 2024b) still lack effective perception of 3D spatial relations and precise location generation, which is a fundamental capability for spatial reasoning and generation tasks. In 3D scenes, spatial information naturally exists at different levels, including that of individual objects, object groupings, and entire architectures etc. Consequently, the majority of existing 3D MLLMs depend on holistic 3D scene information or specifically designated regions for 3D vision-language(3D VL) tasks, failing to capture multi-level location-based information.

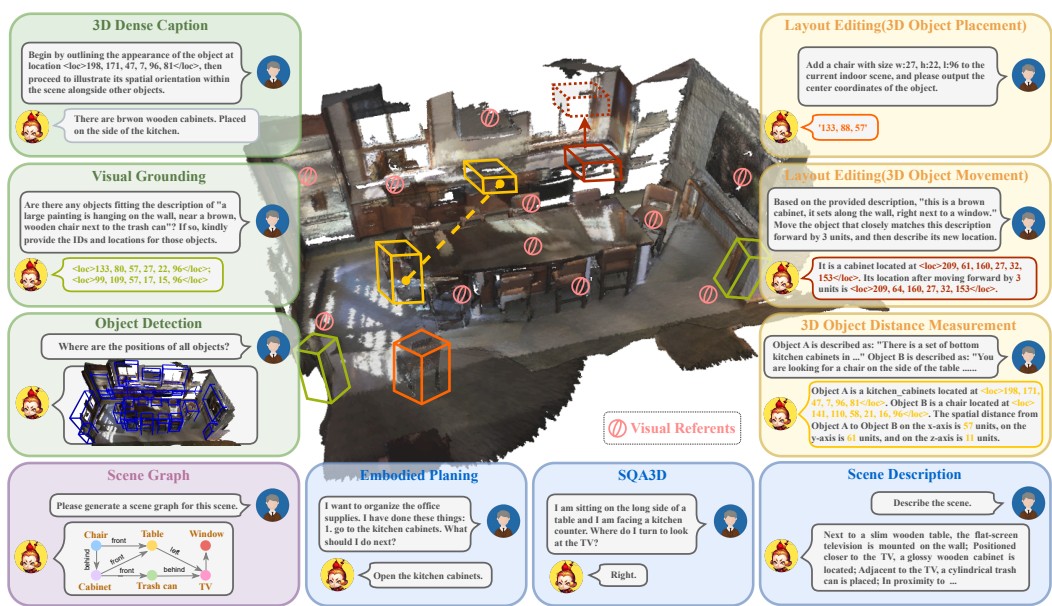

Figure 1: **High-level overview of Spatial 3D-LLM.** It is a 3D MLLM dedicated to improving the capabilities of 3D spatial perception and reasoning by enriching the spatial embeddings of 3D scenes, performing well on various 3D vision-language tasks.

Considering spatial intelligence from the perspective of tasks and datasets, several works(Cheng et al., 2024; Chen et al., 2024a; Cai et al., 2024) have improved image-based spatial reasoning capabilities by generating large-scale spatially-aware training data. They hypothesize that VLMs' limited spatial reasoning capability is due to the lack of 3D spatial knowledge in training data(Cheng et al., 2024; Chen et al., 2024a). Those generated question answering datasets are mainly related to estimating object pair relationships and metric measurements. Existing 3D instruction following datasets(Li et al., 2023; Yang et al., 2024a; Lyu et al., 2024) support a variety of spatial tasks, including visual question answering, visual grounding, and spatial relationships estimation. However, these datasets mainly concentrate on perceiving coarse-grained and abstract object relationships while leaving fine-grained measurement unexplored. Moreover, they typically focus on local object interactions, neglecting the utilization of commonsense knowledge of object-scene spatial information.

In light of the mentioned deficiencies in existing 3D instruction datasets, we propose two novel tasks, namely 3D object distance measurement and 3D layout editing in 3D scenes, to evaluate the spatial perception capabilities of 3D MLLMs. We construct a 3D instruction dataset called **M**easure **O**bject **D**istance and **L**ayout **E**diting (MODLE) that is furnished with 263K vision-language annotations specifically targeted towards these tasks. Inferring precise distances between objects enhances fine-grained spatial perception, while performing object placement and movement in a 3D scene fosters a deeper understanding of object-scene spatial information, accumulating commonsense knowledge for downstream tasks. The agent will gain wider and deeper spatial awareness and be better equipped to interact within complex 3D environments. By successfully completing these two tasks, the spatial intelligence of the agents can be significantly enhanced.

Given the aforementioned concerns regarding the inadequate exploitation of spatial information in existing 3D MLLMs, we propose **Spatial 3D-LLM**, a 3D MLLM aimed at improving capabilities of spatial perception and reasoning for 3D VL tasks by enriching the spatial embeddings of 3D scenes, as depicted in Figure 1. Spatial 3D-LLM incorporates a frozen 3D scene encoder, an LLM backbone, and a meticulously designed progressive spatial awareness scheme that includes intra-referent clustering and abstraction, inter-referent message passing, and contextual referent-scene interactions. This spatial awareness visual referent evolution begins with relation-based clustering. It then continues with inter-referent message passing to model spatial distribution based on the distances between different referents. Finally, it encompasses broader contextual information by considering the interactions between referents and the surrounding environment. This stepwise scheme progressively

captures spatial information as the perception field expands, injecting location-enriched spatial knowledge into the 3D scene embeddings. These enhanced embeddings serve as visual prompt for end-to-end instruction tuning, eliminating the need for task-specific optimizations. Additionally, by applying our progressive spatial awareness scheme and our proposed dataset **MODEL**, the LLM could capture both fine-grained spatial information and commonsense knowledge. This further strengthens spatial awareness and improves overall task performance.

In summary, our contributions are as follows:

- We propose two novel location-related tasks in 3D scenes, namely 3D object distance measurement and 3D layout editing. We construct **MODLE**, a 3D instruction dataset furnished with 263K vision-language annotations towards these tasks. Fine-grained spatial perception and commonsense knowledge of object-scene spatial relationships can be significantly enhanced through the tasks.
- We present **Spatial 3D-LLM**, a 3D MLLM that improves 3D spatial perception and reasoning capabilities by enriching the spatial embeddings of 3D scenes. Spatial 3D-LLM features a progressive spatial awareness scheme that captures spatial information as the perception field expands, injecting location-enriched spatial knowledge into the 3D scene embeddings.
- Experimental results demonstrate that our method achieves state-of-the-art performance across diverse 3D VL tasks, especially those concerning locations and spatial relationships. This reveals the effectiveness of our progressive spatial awareness scheme for mining enhanced spatial information and the usage of commonsense knowledge derived from the MODEL dataset.

## 2 RELATED WORK

### 2.1 SPATIAL INTELLIGENCE IN 3D VISION-LANGUAGE TASKS

Diverse 3D VL tasks pose disparate demands on a model's capability of spatial perception and reasoning within 3D environments. For instance, **3D Visual Question Answering (3D-VQA)** (Azuma et al., 2022; Ye et al., 2021; Zhao et al., 2022; Ma et al., 2022) primarily rely on understanding the holistic scene to provide answers or descriptions, without delving deeply into object-to-object spatial configurations. **3D Visual Grounding(3D-VG)** (Achlioptas et al., 2020; Chen et al., 2020) and **3D Object Detection** (Qi et al., 2019; Lin et al., 2013) demand precise spatial localization, focusing on identifying and locating specific objects or regions within the 3D space. Additionally, **3D Dense Captioning** (Chen et al., 2023a; 2021) involves generating detailed descriptions for various regions or objects in a 3D scene, requiring a strong grasp of how objects are positioned and interact within their spatial context.

Existing 3D VL tasks primarily focus on perceiving coarse-grained and abstract object relationships, coupled with concentrating on local object interactions. Our newly proposed tasks, namely 3D object distance measurement and 3D layout editing, enhance fine-grained spatial perception and accumulate commonsense knowledge for downstream tasks, advancing the capability of spatial intelligence.

### 2.2 SPATIAL LEARNING IN 3D MULTIMODAL LLMS

Recent advancements in 3D MLLMs(Hong et al., 2023; Li et al., 2024; Chen et al., 2023b; Huang et al., 2023; Chen et al., 2024b) have explored a variety of spatial learning paradigms. These architectures typically comprise 3D vision perceptrons, projectors, and LLM backbones. 3DLLM(Hong et al., 2023) introduced location special tokens to better capture 3D spatial information, enabling models to output 3D coordinates. LL3DA(Chen et al., 2024b) used clicks and boxes as visual prompts to interact with 3D embeddings and generate spatial queries. SpatialRGPT(Cheng et al., 2024) enhanced region-level spatial reasoning in VLMs by improving regional information representation and spatial knowledge acquisition. Chat-3D v2(Huang et al., 2023) segmented scenes into objects, mapped each with an index, and used special tokens to capture 3D attributes and spatial relations. Grounded 3DLLM(Chen et al., 2024d) introduced special noun phrase tokens to reference 3D scenes and let models process 3D-textual data sequences. Most existing 3D MLLMs rely on holistic 3D scene information or specifically designated regions, missing multi-level location-based information.

Distinguished from current approaches, our method explores a progressive spatial awareness scheme that incorporates intra-referent clustering and abstraction, inter-referent message passing, and contextual referent-scene interactions, injecting richer spatial knowledge into the 3D scene embeddings.

Table 1: **Statistic results of our proposed MODLE dataset.** [BOX] represents the 3D bounding box of an object, and [DIS] represents the distance value between objects.

| Tasks | #3D Scan | #Language | | Object-level | Text Instructions | Output Type |
|---|---|---|---|---|---|---|
| | | Train | Val | | | |
| 3D Object Distance Measurement | 0.7K | 171K | 2K | Multi | <obj caption> | [BOX], [DIS] |
| 3D Object Movement | 0.7K | 36K | 9K | Single | <obj caption> | [BOX] |
| 3D Object Placement | 0.69K | 34K | 9K | Single | <obj caption> | [BOX] |

## 3 DATASETS

We propose 3D object distance measurement and 3D layout editing tasks for improving 3D spatial perception capabilities of our **Sptial 3D-LLM**, and accumulating commonsense knowledge for downstream tasks. Hence, we construct a visual language instruction dataset for these two tasks, namely **MODLE**. Statistics for the datasets are provided in Table 1, with relevant evaluation metrics and examples are shown in Appendix A.

### 3.1 3D OBJECT DISTANCE MEASUREMENT TASK

This task focuses on inferring 3D spatial distance between two objects within 3D scenes. We create 173K text-location pairs. Questions are made with manually defined templates, generating synthetic data by filling in object descriptions sourced from ScanRefer(Chen et al., 2020) dataset. Answers are derived from the actual 3D bounding box coordinates of the objects. We introduce Interaction Tokens to distinguish between coordinate information and distance values in the output. Coordinates are put within <loc></loc> tokens, and distances are marked with <gap></gap> tokens.

### 3.2 3D LAYOUT EDITING TASK

This task demands the model have 3D layout editing capabilities. We design two subtasks: object movement and placement. Unlike the 3D-VG task that grounds an object in the scene, 3D layout editing requires a precise understanding of 3D spatial positions for predicting new object positions.

For the object movement task, the model is required to relocate an object in the scene based on its description and an editing instruction. We define a template including the object description and movement instructions to construct the dataset with 45K text-location pairs. The object descriptions come from the ScanRefer dataset, and the movement instructions are randomly generated. In the object placement task, the model needs to understand the holistic scene and accurately place an object of a specified size within the scene layout. We created 33K sub-scenes from the ScanNet(Dai et al., 2017) dataset, each with 3 to 8 objects. During training and evaluation, we randomly mask one object from each sub-scene and require the model to predict a reasonable spatial position.

## 4 METHODOLOGY

We propose Spatial 3D-LLM, a 3D MLLM for comprehensive 3D scene understanding, 3D visual grounding, 3D spatial measurement, and 3D scene layout editing. The main pipeline of Spatial 3D-LLM is illustrated in Figure 2. Spatial 3D-LLM incorporates a frozen 3D scene encoder, an LLM backbone, and a meticulously designed progressive spatial awareness scheme that includes intra-referent clustering and abstraction, inter-referent message passing, and contextual referent-scene interactions. Next, we will explain the details of each part.

### 4.1 SCENE ENCODER

To handle the point clouds in the 3D scene, we utilize PointNet++ (Qi et al., 2017) as our scene encoder, which employs a hierarchical neural network to convert the unordered point set into an unordered set of point features. To represent the input 3D scene, the scene encoder outputs 1,024 point tokens, $F_{\text{enc}} = [p_{\text{enc}}, f_{\text{enc}}] \in \mathbb{R}^{1,024 \times (3+256)}$, containing scene features $f_{\text{enc}}$ for 256 dimensions and coordinates $p_{\text{enc}}$ for 3 dimensions.

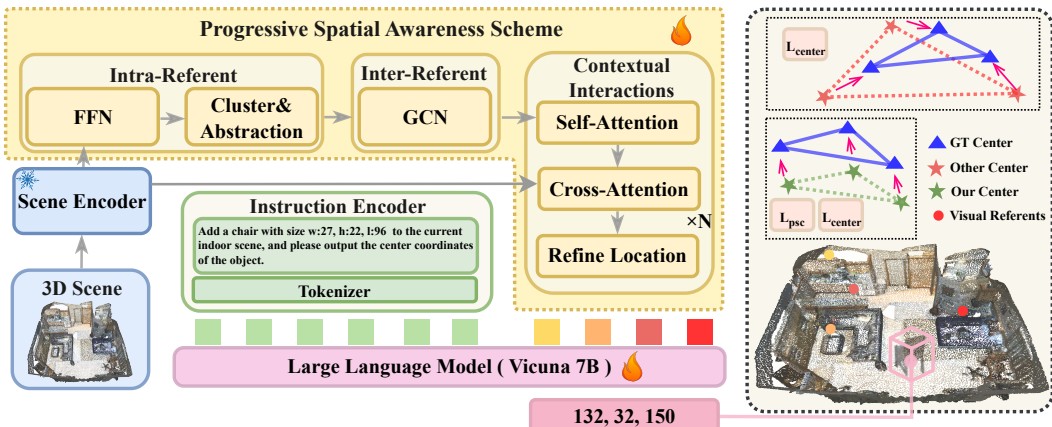

Figure 2: **The model architecture of Spatial 3D-LLM.** It includes a frozen 3D scene encoder, an LLM backbone, and a meticulously designed progressive spatial awareness scheme that incorporates intra-referent clustering and abstraction, inter-referent message passing, and contextual referent-scene interactions, generating location-enriched 3D scene embeddings that serve as visual prompt.

## 4.2 PROGRESSIVE SPATIAL AWARENESS SCHEME

Considering that the majority of existing 3D MLLMs struggle to effectively capture multi-level location-based information, we propose Progressive Spatial Awareness Scheme that encompasses intra-referent clustering and abstraction, inter-referent message passing, and contextual referent-scene interaction. This approach aims to enhance spatial information retrieval as the perception field expands, thereby infusing location-enriched spatial knowledge into the 3D scene embeddings.

### 4.2.1 INTRA-REFERENT

To capture point-to-point relational information within the scene, we propose the Intra-referent module, which comprises Feedforward Neural Network (FFN) layers and a Cluster&Abstraction module. Specifically, we sample 256 points from the encoded set of 1024 scene points using Farthest Point Sampling (FPS), resulting in seed points $F_{\text{seed}} = [p_{\text{seed}}, f_{\text{seed}}] \in \mathbb{R}^{256 \times (3+256)}$, following VoteNet(Qi et al., 2019). Next, 3D spatial offset $\Delta \mathbf{p}_{vote}$ is predicted from the seed point feature $f_{seed}$ by means of FFN layers. With the aim of adjusting seed point location to align with the centers of object, as indicated by:

$$p_{vr} = p_{seed} + FFN(f_{seed}) = p_{seed} + \Delta \mathbf{p}_{vote}$$

we aggregate local information from $F_{\text{seed}}$ for $p_{vr}$ with a Cluster & Abstraction module, to form the visual referent reprention, as $F_{\text{vr}} = [p_{\text{vr}}, f_{\text{vr}}]$. For each visual referent location $p_{vr}$, its neighboring points are grouped to form local regions, and the features of these regions from $f_{enc}$ are abstracted by pooling, mapping the set of points to a feature vector, as visual referent embedding $f_{vr}$:

$$f_{vr} = Pooling(Cluster[p_{vr}, f_{enc}])$$

This process generates visual referent representation $F_{\text{vr}} = [p_{\text{vr}}, f_{\text{vr}}] \in \mathbb{R}^{256 \times (3+256)}$, which encapsulates the internal point-to-point relationships within the local region.

### 4.2.2 INTER-REFERENT

We contend that relying solely on the feature information from local region is insufficient for adequately representing 3D scenes. Consequently, we propose the Inter-Referent module, which employs a Graph Convolutional Network (GCN) model for message propagation to facilitate the modeling of global spatial distribution among visual referents, particularly focusing on the implicit relationships between these referents within the scene.

In this approach, we model the graph nodes using $f_{\text{vr}}$, with edges defined based on the distances between visual referents locations $p_{\text{vr}}$. The forward propagation for each layer of the GCN can be expressed as follows: $H^{(l+1)} = \sigma\left(AH^{(l)}W^{(l)}\right)$, where $H^{(l)}$ represents the node embeddings at layer $l$, $A$ is the adjacency matrix capturing the spatial connections between nodes, $W^{(l)}$ is the weight matrix of the GCN layer, and $\sigma$ is the activation function.

By iteratively training the GCN, we learn an enhanced representation for each visual referent that captures both its local features and the global spatial context from its neighboring visual referents. The output of the layers of GCN is a refined visual referent representation $F_{\text{vr}} = [p_{\text{vr}}, f_{\text{vr}}] \in \mathbb{R}^{256 \times (3+256)}$, which is obtained through inter-referent message passing.

### 4.2.3 CONTEXTUAL INTERACTIONS

To achieve contextual interactions among visual referents and the global scene, we introduce the Context Interactions module, which employs multiple blocks of self-attention, cross-attention, and Refine-Location Module to obtain spatially aware representations of referents. In detail, visual referent representation processed by Inter-referent Module $F_{\text{vr}}$ and the scene features $f_{\text{enc}}$ undergo self-attention and cross-attention layers. The updated $F_{\text{vr}}$ can be claimed as a scene-aware visual referent representation, effectively incorporating object-specific features and spatial positional information, thereby enhancing its comprehensiveness and achieving spatial scene awareness.

**Refine-Location Module.** To further improve the precision of visual referent location predictions, we introduce the *Refine-Location Module*, designed to refine the spatial positioning of referents by minimizing the relative distance to their ground truth coordinates.

This module comprises multiple layers of FFN that learn positional offsets to adjust the locations of visual referents, aligning them more closely with the object's coordinate center. We define a visual referent's ground truth location as the centroid of the nearest object. Consequently, supervision derives from these ground truth locations, aiming to minimize both center distance and pairwise distance between predicted and actual visual referent pairs, quantified through *center loss* ($\mathcal{L}_{\text{center}}$) and the *pairwise spatial constraint loss* ($\mathcal{L}_{\text{psc}}$) which are computed as:

$$\mathcal{L}_{\text{center}} = \frac{1}{M}\sum_{i=1}^{M} \|q_{\text{vr}}^{(i)} - q_{\text{gt}}^{(i)}\|_2, \quad \mathcal{L}_{\text{psc}} = \frac{1}{N}\sum_{i=1,j=1}^{N} \|k_{\text{vr}}^{(ij)} - k_{\text{gt}}^{(ij)}\|_2,$$

where $M$ is the number of visual referent, $N$ is the number of visual referent pairs, $q_{\text{vr}}^{(i)}$ and $q_{\text{gt}}^{(i)}$ are denoted as the coordinates of the predicted and ground truth visual referent, $k_{\text{vr}}^{(ij)}$ is the predicted distance of visual referent pairs $(i, j)$, $k_{\text{gt}}^{(ij)}$ is the corresponding ground truth distance. This loss penalizes the Euclidean distance between the predicted and ground truth distance, encouraging the model to predict more accurate visual referent locations.

By using a progressive visual referent evolution approach that enhances the perception field with spatial information, learned 3D scene embeddings effectively capture location-enriched spatial knowledge. This allows our model to excel in spatial position perception and improve its ability for spatial understanding and reasoning in 3D vision-language tasks.

### 4.3 SPATIAL 3D-LLM TRAINING

For representing 3D point coordinates occuring the text, following 3D-VLA(Zhen et al., 2024) and LL3DA(Chen et al., 2024b), we normalize the point cloud coordinates into discrete unsigned integers within the range [0-255]. This representation is distinguished by special token `<loc></loc>`, which help differentiate the spatial coordinates from other data.

To integrate the visual prompt, denoted as $F_{visual}$, which contains both visual referent features $f_{vr}$ and location representation $p_{vr}$ into the large language model (LLM). We introduce a trainable projector consist multi-layers of FFN to align $F_{visual} = [f_{vr}, p_{vr}]$ as **Visual Prompt** within the language space of LLM, allowing the model to process 3D spatial information alongside natural language input.

We use the instruction tuning paradigm for training our Spatial 3D-LLM. With VL understanding and VL grounding tasks consist of the training dataset, we get the loss, denoted as $\mathcal{L}_{\text{LLM}}$, is computed based on the model's performance on these tasks. In addition to $\mathcal{L}_{\text{LLM}}$, we also introduce $\mathcal{L}_{\text{psc}}, \mathcal{L}_{\text{center}}$ to get more precise coordinates generation and understanding. Thus, the overall optimization objective is the sum of these two losses:

$$\mathcal{L}_{\text{total}} = \mathcal{L}_{\text{LLM}} + \alpha_1 \mathcal{L}_{\text{psc}} + \alpha_2 \mathcal{L}_{\text{center}},$$

where $\mathcal{L}_{\text{LLM}}$ is the loss associated with the instruction tuning tasks for our based LLM, $\alpha_1$ and $\alpha_2$ are weighting factors. By optimizing this combined loss, Spatial 3D-LLM learns both precise spatial information through the spatial loss and instruction-following capabilities via our based LLM instruction tuning loss.

## 5 EXPERIMENTS

### 5.1 DATASETS AND IMPLEMENTATION DETAILS

**Datasets.** To evaluate the performance of our model, we require 3D scene point clouds along with a visual-language task dataset. For the 3D scene input, we utilize ScanNet(Dai et al., 2017), a real 3D indoor scene dataset that includes 1,201 training scenes and 312 testing scenes. For visual-language data, we incorporate Scan2Cap(Chen et al., 2021), ScanQA(Azuma et al., 2022), SQA3D(Ma et al., 2022), and embodiedQA(Hong et al., 2023) from 3D-LLM for training and evaluation of visual-language understanding tasks. Additionally, we use ScanRefer(Chen et al., 2020) and Multi3DRefer(Zhang et al., 2023) for single- and multi-object grounding, and leverage proposed distance measurements, object movement, and object placement tasks for precise spatial position perception and generation.

**Implementation Details.** We initialize the weights of the 3D scene encoder using the pre-trained Vote2Cap-DETR(Chen et al., 2024c). The large language model utilizes the pre-trained Vicuna-7B and implement LoRA for instruction-tuning. During the training process, we jointly train the progressive spatial awareness scheme and the LoRA parameters across all task datasets. We employ AdamW as the optimizer, with a learning rate between $10^{-4}$ and $10^{-7}$ and a weight decay of 0.1. All experiments are conducted on eight A100 GPUs within one day.

### 5.2 COMPARISON WITH SOTA MODELS

To evaluate the capabilities of our model, we present the evaluation results on two types of tasks: 3D vision-language understanding tasks and 3D vision-language grounding tasks. The qualitative results are shown in Figure 3. It is worth noting that all tasks were trained together during the training process, and for each evaluation task, the evaluation metrics come from the same model weight.

#### 5.2.1 3D VISION-LANGUAGE UNDERSTANDING

We assess the model's ability to understand 3D scenes through the Scan2Cap, ScanQA and SQA3D tasks, with Table 2 reporting the explicit performance on each task. We categorize the existing methods into three groups: task-specific models tailored for downstream tasks; task-specific fine-tuned approaches that involve pretraining a unified 3D backbone followed by subsequent fine-tuning for specific tasks; and generalist models capable of comprehending a range of 3D vision-language tasks.

**Analysis** Table 2 shows that our method surpasses most methods in terms of CIDEr(Vedantam et al., 2015), BLEU-4(Papineni et al., 2002), METEOR(Banerjee & Lavie, 2005), and ROUGE(Lin, 2004) across all three tasks. For example, in the Scan2Cap task, which requires a model to localize and generate descriptive captions for any object in a 3D scene. In the Scan2Cap task, which involves localizing and generating descriptive captions for objects in 3D scenes, our method achieves a significantly higher CIDEr score, reflecting its ability to generate more accurate and contextually relevant captions. Similarly, in the ScanQA task, which tests the model's ability to answer questions with more semantic diversity about 3D scenes, our method shows notable improvements across all metrics, particularly in CIDEr and BLEU-4. Furthermore, in the SQA3D task, which involves

Table 2: **Quantitative comparison with SOTA models on 3D VL understanding tasks.** "C" stands for "CIDEr", "B-4" for "BLEU-4", "M" for "METEOR", "R" for "ROUGE", and "EM@1" for top-1 exact match. The n-gram metrics for Scan2Cap are governed by IoU@0.5. $^\dagger$ indicates answering questions via prompting GPT-3 with the generated scene caption.

| | Scan2Cap | | | | ScanQA | | | | SQA3D |
|---|---|---|---|---|---|---|---|---|---|
| | C | B-4 | M | R | C | B-4 | M | R | EM@1 |
| *Task-specific models* | | | | | | | | | |
| Scan2Cap(Chen et al., 2021) | 35.2 | 22.4 | 21.4 | 43.5 | - | - | - | - | 41.0$^\dagger$ |
| Vote2Cap-DETR(Chen et al., 2023a) | 61.8 | 34.5 | 26.2 | 54.4 | - | - | - | - | - |
| ScanRefer+MCAN(Chen et al., 2020) | - | - | - | - | 55.4 | 7.9 | 11.5 | 30.0 | - |
| ScanQA(Azuma et al., 2022) | - | - | - | - | 64.9 | 10.1 | 13.1 | 33.3 | 47.2 |
| *Task-specific fine-tuned* | | | | | | | | | |
| 3D-VisTA(Zhu et al., 2023) | 66.9 | 34.0 | 27.1 | 54.3 | 69.6 | 10.4 | 13.9 | 35.7 | 48.5 |
| 3D-LLM (FlanT5)(Hong et al., 2023) | - | - | - | - | 69.4 | 12.0 | 14.5 | 35.7 | |
| Chat-3D v2(Huang et al., 2023) | - | - | - | - | 77.1 | 7.3 | 16.1 | 40.1 | - |
| LL3DA(Chen et al., 2024b) | 65.2 | 36.8 | 26.0 | 55.1 | 76.8 | 13.5 | 15.9 | 37.3 | |
| *Generalist models* | | | | | | | | | |
| LL3DA(Chen et al., 2024b) | 63.0 | 36.0 | 25.7 | 54.7 | 75.7 | 13.3 | 15.4 | 37.0 | - |
| Grounded 3D-LLM(Chen et al., 2024d) | 70.6 | 35.5 | - | - | 72.7 | 13.4 | - | - | - |
| Spatial 3D-LLM (Ours) | **72.2** | 34.6 | 23.1 | 54.3 | **82.5** | **13.9** | **16.8** | 39.1 | 46.2 |

Table 3: **Quantitative comparison with SOTA models on 3D VL grounding tasks.** [BOX] indicates models that output 3D bounding boxes, while [ID] indicates models that output individual object IDs. ReGround3D 3D-LLM refers to the reproduced 3D-LLM results from the ReGround3D model.

| | Output Type | ScanRefer Grd. | | Multi3DRef Grd. | |
|---|---|---|---|---|---|
| | | Acc@0.25 | Acc@0.5 | F1@0.25 | F1@0.5 |
| ScanRefer(Chen et al., 2020) | [BOX] | 37.3 | 24.3 | - | - |
| M3DRef-CLIP(Zhang et al., 2023) | [BOX] | 51.9 | 44.7 | 42.8 | 38.4 |
| LLM-Grounder(Yang et al., 2023) | [BOX] | 17.1 | 5.3 | - | - |
| Chat-3D v2(Huang et al., 2023) | [ID] | 35.9 | 30.4 | - | - |
| ReGround3D 3D-LLM(Zhu et al., 2024) | [BOX] | 33.1 | 28.7 | - | - |
| Grounded 3D-LLM(Chen et al., 2024d) | [ID] | 47.9 | 44.1 | 45.2 | 40.6 |
| Spatial 3D-LLM (Ours) | [BOX] | 44.3 | 37.2 | **48.3** | **41.2** |

answering situated questions in complex 3D environments, Spatial 3D-LLM once again excels, showcasing its robustness in understanding both spatial and linguistic nuances. Overall, our method consistently surpasses other models in key performance metrics, demonstrating its advanced spatial reasoning capabilities and comprehensive understanding of 3D scenes.

### 5.2.2 3D VISION-LANGUAGE GROUNDING

Table 3 presents a quantitative comparison between our method and several SOTA models on 3D vision-language grounding tasks, evaluated on the ScanRefer(Chen et al., 2020) and Multi3DRef(Zhang et al., 2023) benchmarks. We report the evaluation metrics of Acc@0.25 and Acc@0.5 for visual grounding on ScanRefer, and F1@0.25 and F1@0.5 for multi-object visual grounding on Multi3DRef.

**Analysis** Table 3 shows that our method demonstrates competitive performance across both tasks. Specifically, in the ScanRefer visual grounding task, our approach achieves the Acc@0.25 score of 44.3% and Acc@0.5 of 37.2%, closely matching the performance of Grounded 3D-LLM and outperforming other several baselines. In the Multi3DRef visual grounding task, our model achieves the F1@0.5 score of 41.2% and F1@0.25 score of 48.3%, which outperforms than other baselines such as ReGround3D and Grounded 3D-LLM.

These results demonstrate the effectiveness of our method, particularly in multi-object grounding scenarios. While our model slightly lags behind the top-performing models in terms of overall

Table 4: **Ablation studies of different tasks.** **U** refer to train on 3D VL understanding tasks, **G** refer to training on 3D VL grounded tasks, and **O** refer to training on our proposed task.

| | Scan2Cap | | Multi3DRef Grd | | Movement | Placement | Measurement |
|---|---|---|---|---|---|---|---|
| | C | B-4 | F1@0.25 | F1@0.5 | Acc@0.5 | Acc@0.5 | X/Y/Z-mARE@0.5 |
| Ours (U only) | 62.9 | 30.4 | - | - | - | - | - |
| Ours (G only) | - | - | 44.6 | 38.3 | - | - | - |
| Ours (U + G) | 67.7 | 32.1 | 47.2 | 39.8 | - | - | - |
| Ours (G + O) | - | - | 45.4 | 38.6 | 37.6 | 63.2 | 2.3/1.7/3.4 |
| Ours (U + G + O) | 72.2 | 34.6 | 48.3 | 41.2 | 40.3 | 66.4 | 2.0/1.4/2.4 |

accuracy in the ScanRefer task, it excels in the Multi3DRef task, showing its strength in handling complex spatial relationships across multiple objects. The consistent performance across different metrics highlights the robustness and versatility of our approach in 3D Vision-Language grounding tasks. Notably, our model directly outputs precise 3D bounding boxes for object localization, offering a significant advantage over similar previous SOTA methods like ReGround3D 3D-LLM.

### 5.3 ABLATION STUDIES

To further evaluate the effectiveness of the joint training of the 3D VL understanding task and the 3D VL grounding task, as well as the implementation of progressive spatial awareness scheme in enhancing the performance of our Spatial 3D-LLM, we evaluate our model on both existing tasks and proposed benchmark and conduct ablation studies.

**Analysis of ablation studies on different tasks.**

Table 4 demonstrates the performance across different training setups: 3D VL understanding (**U**), 3D VL grounding (**G**), and our proposed task (**O**). Training on all tasks (U + G + O) yields the best overall performance. For Scan2Cap task, it slightly improves BLEU-4 while maintaining a strong CIDEr score. In the Multi3DRef task, (U + G + O) outperforms G-only with higher score of F1@0.25 and F1@0.5. Similarly, for object movement and placement tasks, (U + G + O) achieves higher accuracy compared to G-only. Finally, for the 3D object distance measurement task, (U + G + O) significantly reduces the mean absolute relative error, demonstrating the importance of combining all tasks for effective spatial reasoning. Overall, these findings emphasize the importance of joint training in improving 3D scene understanding and spatial perception.

**Analysis of ablation studies of different components.**

Table 5 presents the results of ablation studies on key components: Intra-Referent(**C1**), Inter-Referent(**C2**) and Contextual Interactions(**C3**). Our model (C1 + C2) outperforms C1 alone, demonstrating the effectiveness of the Inter-Referent Module(**C2**). This is attributed to its ability to learn the implicit relationships between visual referents. Our full model (C1 + C2 + C3) consistently outperforms both the C1 model and the (C1 + C2) model, demonstrating the effectiveness of the Contextual Interaction module (**C3**) in learning referent-scene interactions. In Scan2Cap, it achieves the highest CIDEr and BLEU-4 scores. Similarly, for Multi3DRef, our model outperforms the C1 and C2 variants. The full model also shows superior performance in object editing tasks, achieving higher accuracy in both movement and placement tasks. For the measurement task, our model demonstrates lower mean absolute relative error (X/Y/Z-mARE) compared to the alternatives, emphasizing the contribution of each component to spatial understanding. Furthermore, our proposed progressive spatial awareness scheme has a significant impact on performance, providing superior feature extraction that supports complex spatial reasoning.

### 5.4 LIMITATION

The key limitation of our Spatial 3D-LLM is the sophisticated computational processes at the core of our framework, such as the Progressive Spatial Awareness Scheme requires significant computational resources. This intensity may limit the model's applicability in real-time applications, especially in environments with limited computing power. To address this, advancements in both hardware and software technologies are necessary. We believe that enhancing the computational power of

Table 5: **Ablation studies of different components.C1** represents Intra-Referent module, **C2** represents Inter-Referent module, and **C3** represents Contextual Interactions module.

| C1 | C2 | C3 | Scan2Cap | | Multi3DRef Grd | | Movement | Placement | Meassurment |
|----|----|----|----|----|----|----|----|----|----|
| | | | C | B-4 | F1@0.25 | F1@0.5 | Acc@0.5 | Acc@0.5 | X/Y/Z-mARE@0.5 |
| ✓ | | | 52.1 | 32.5 | 30.4 | 15.9 | 31.7 | 46.8 | 7.5/8.7/7.2 |
| ✓ | ✓ | | 67.1 | 33.3 | 43.4 | 37.0 | 34.2 | 59.7 | 2.6/2.1/4.5 |
| ✓ | ✓ | ✓ | 72.2 | 34.6 | 48.3 | 41.2 | 40.3 | 66.4 | 2.0/1.4/2.4 |

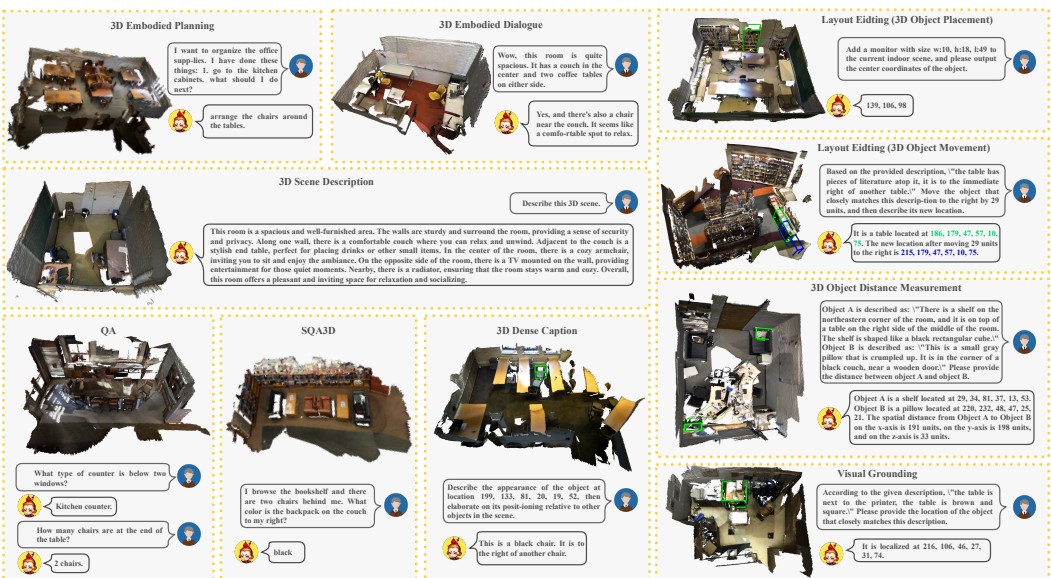

Figure 3: **Qualitative Results.** We provide several visualization results on various 3D vision and language tasks.

terminal equipment and employing model quantization techniques could potentially mitigate some of the barriers to the practical implementation of Spatial 3D-LLM.

## 6 CONCLUSION

In this paper, we have presented Spatial 3D-LLM, a multi-modal LLM for 3D scene understanding and spatial perception, which could fully exploit the spatial information within 3D scenes. By carefully designing a progressive spatial awareness scheme within our framework, Spatial 3D-LLM achieves superior performance across a spectrum of 3D VL tasks. Moreover, two novel tasks including 3D object distance measurement and layout editing are proposed to enhance fine-grained spatial perception and commonsense knowledge. The experimental results verify our Spatial 3D-LLM's capability in 3D scene understanding and spatial perception. Our future work includes expanding the diversity of the training datasets to encompass more complex and varied 3D scenes, which would enhance the model's generalizability. Additionally, we aim to investigate methods to improve real-time performance without compromising accuracy for practical applications such as robotics and augmented reality.

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

# A  APPENDIX

## A.1  DATASET CASES

**The case of 3D object distance measurement task**

- **Scene:** `scene0011_00`
- **Question:** *"Object A is described as: 'There is a set of bottom kitchen cabinets in the room. It has a microwave in the middle of it.' Object B is described as: 'You are looking for a chair on the side of the table facing the ovens. It will be the chair near the rail.' Please provide the distance between Object A and Object B."*

- **Answer:** *"Object A is a kitchen_cabinets located at `<loc>198, 171, 47, 7, 96, 81</loc>`. Object B is a chair located at `<loc>141, 110, 58, 21, 16, 96</loc>`. The spatial distance from Object A to Object B on the x-axis is `<gap>57</gap>` units, on the y-axis is `<gap>61</gap>` units, and on the z-axis is `<gap>11</gap>` units."*

**The cases of 3D layout editing task**

- **Object Movement - Scene:** `scene0011_00`
- **Question:** *"Based on the provided description, 'this is a brown cabinet, it sets along the wall, right next to a window.' Move the object that closely matches this description forward by 110 units, and then describe its new location."*
- **Answer:** *"It is a cabinet located at `<loc>209, 61, 160, 27, 32, 153</loc>`. Its location after moving forward by 110 units is `<loc>209, 171, 160, 27, 32, 153</loc>`."*

- **Object Placement - Scene:** `scene0011_00`
- **Question:** *"Add a chair with size w:27, h:22, l:96 to the current indoor scene, and please output the center coordinates of the object."*
- **Answer:** *"133, 80, 57"*

## A.2 EVALUATION METRICS OF OUR CONSTRUCTED TASKS

**3D object distance measurement task** To assess the accuracy of distance predictions, we draw inspiration from the evaluation of 3D-VG, focusing on the localization accuracy of Objects A and B. We follow the approach used in Chen et al. (2010), employing **absolute relative error (ARE)** to evaluate distance prediction accuracy. Our primary metric, **mARE@kIOU**, measures the mean absolute relative error for predictions on the X, Y, and Z axes, providing a detailed assessment of the model's spatial reasoning capabilities.

**3D layout editing task** To evaluate the accuracy of object editing in the scene, we follow the metrics used in the 3D-VG task, calculating the Intersection over Union (IoU) between the predicted bounding box and the ground truth to assess the rationality of the predicted positions.

