# OpenReview forum: "Spatial 3D-LLM: Progressive Spatial Awareness for Advanced 3D Vision-Language Understanding"
_ICLR.cc/2025/Conference — ICLR 2025 Conference Withdrawn Submission_

### Official Review · Reviewer_CnqC · 2024-11-03

**Soundness:** 3
**Presentation:** 2
**Contribution:** 2
**Rating:** 5
**Confidence:** 5

**Summary:**

The paper focuses on 3D-LLM and designs an architecture to improve spatial awareness. In addition to the architecture design, the paper also studies two new tasks on 3D-LLM, including object distance measurement and layout editing.

**Strengths:**

Current 3D LLMs still struggle to capture rich spatial information. The paper proposes an architectural improvement to address this issue. Additionally, the newly introduced tasks of distance measurement and object movement have great potential for valuable applications.

**Weaknesses:**

My main concern is twofold. First, the method lacks an important comparison with LEO (An Embodied Generalist Agent in 3D World, ICML'24). The reported performance on public benchmarks seems to underperform LEO’s results, and this discrepancy is not sufficiently explained. Second, the newly introduced task does not include important baseline comparisons. For the measurement and movement task, incorporating several simple approaches as meaningful baselines would enhance the evaluation. For example, using off-the-shelf 3D segmentation and detection to extract object bounding boxes for direct measurement or constructing scene graphs from 3D scenes and using existing LLMs for both measurement and movement tasks.

**Questions:**

* How does Spatial 3D-LLM compared to LEO?
* How to obtain the centers of the object for the refinement loss? Does the number of object center points need to be the same as seed points (256)?
* Is it possible to use off-the-shelf 3D segmentation and detection to extract object bounding boxes for direct measurement? Or construct scene graphs and leverage LLMs for both measuring and movement tasks?
* Does the movement task also consider whether there is space for the movement? Or is it more like an object localization task where the model only needs to ground the described object’s location and apply the movement?
* How do other baseline methods perform on distance measurement/movement tasks if fine-tuned with the proposed new data?

---

### Official Review · Reviewer_YVbz · 2024-11-04

**Soundness:** 2
**Presentation:** 3
**Contribution:** 2
**Rating:** 3
**Confidence:** 5

**Summary:**

This work addresses the task of developing 3D-LLM for scene understanding. To enhance fine-grained spatial perception and commonsense knowledge of object-scene spatial relationships, it proposes two types of training instructions: 3D object distance measurement and 3D layout editing. This work also introduces a progressive spatial awareness scheme to connect point cloud feature and the LLM.

**Strengths:**

1. Writing is mostly clear.
2. Most of the important baselines are comprehensively compared.
3. Ablations on the proposed tasks and components are appreciated.

**Weaknesses:**

1. The motivation for introducing object distance measurement and layout editing data is unclear. These data are generated by filling a template with the ScanRefer dataset, and the key to solving these tasks largely relies on 3D visual grounding from ScanRefer (plus simple integer addition or subtraction). I question the necessity of further processing the existing ScanRefer dataset to create these data, as such processing renders the contributions of these two tasks incremental.
2. From Table-4, including the two proposed data types only marginally improves performance, raising concerns about their benefits.
3. For the ablation study, it would be more informative and convincing to show performance on existing benchmarks such as ScanRefer, ScanQA, and SQA3D, rather than only the new tasks. Specifically, I am interested in seeing the effect of training data (tasks) on ScanRefer in the ablations.
4. I would like the authors to provide more qualitative results. Specifically, please run the models for the ScanNet validation set on scenes scene0568_00, scene0169_00, and scene0300_00 with the following prompts:
- “Describe the scene in detail.”
- “List the objects and their quantities in the scene.”
Additionally, for scene0169_00, please prompt the model with: “Find the red backpack” and visualize its location.
5. Missing discussions of object-level and point-based 3D LLM like PointLLM, ShapeLLM, MiniGPT-3D, and token-based 3D localization models like SceneScript.
6. I would like to note that Chat-3D-v2 has an updated version (Chat-Scene, NeurIPS 2024), which demonstrates much better performance. While a comparison with this new version may not be necessary, both the authors and other reviewers should be aware of the performance gap. I suggest that the authors highlight the fact that the proposed method does not require an off-the-shelf 3D detector to pre-detect objects in the scene. This is a significant difference from some of the baselines (including Chat-Scene), and direct comparisons may be unfair, as models with pre-detected objects typically show better results.
7. I am willing to raise my score if the authors can address most of my concerns.

**Questions:**

See weaknesses.

---

### Official Review · Reviewer_T5Q9 · 2024-11-08

**Soundness:** 3
**Presentation:** 3
**Contribution:** 3
**Rating:** 5
**Confidence:** 4

**Summary:**

This paper aims to enhance existing LLMs' spatial perception and reasoning capabilities for 3D vision-language tasks. To achieve this, the authors propose Spatial 3D-LLM which incorporates an LLM backbone and a meticulously designed progressive spatial
awareness scheme to capture the spatial information that serves as the visual prompt. Additionally, they propose a 3D instruction dataset with 263K vision-language annotations called MODEL for two novel location-related tasks: 3D object distance measurement and 3D layout editing.

**Strengths:**

- The proposed approach achieves the SOTA results on several benchmarks on the tasks of 3D visual grounding and understanding.

- The explanation of the methodology is easy to follow and understand.

**Weaknesses:**

- Lack of some details about the proposed METHOD dataset, like the annotation pipeline.

- Some experiment results are missing. Only ablation studies on the METHOD dataset are provided to show the effectiveness of different components. However, I'm more curious about the performance of the existing 3D VLMs on the task of object distance measurement and layout editing. I think this comparison experiment is quite important to verify your motivation.

- Some notations are confusing. In the method section, the visual referent representation after the Intra-Referent module is named $F_{vr}$,  while the feature after the Inter-Referent module is also called $F_{vr}$. I suggest the author use a different notation for the feature after the Inter-Referent module.

**Questions:**

- How to determine the $q_{gt}$ and $k_{gt}$ used in $L_{center}$ and $L_{psc}$.

-  The number of objects can be different for different scenes, however, from my understanding, the number of $q_{vr}$ is equal to the number of visual referents which is 256 in your experiment. Then, how to conduct the $L_{center}$ and $L_{psc}$.

---

### Note · Authors · 2024-11-26

I have read and agree with the venue's withdrawal policy on behalf of myself and my co-authors.